# Pulsatile contractions and pattern formation in excitable actomyosin cortex

**Michael F. Staddon**[1,2,3], **Edwin M. Munro**[4,5], **Shiladitya Banerjee**[6]*

**1** Center for Systems Biology Dresden, Dresden, Germany, **2** Max Planck Institute for the Physics of Complex Systems, Dresden, Germany, **3** Max Planck Institute for Molecular Cell Biology and Genetics, Dresden, Germany, **4** Department of Molecular Genetics and Cell Biology, University of Chicago, Chicago, Illinois, United States of America, **5** Institute for Biophysical Dynamics, University of Chicago, Chicago, Illinois, United States of America, **6** Department of Physics, Carnegie Mellon University, Pittsburgh, Pennsylvania, United States of America

* shiladtb@andrew.cmu.edu

**Data Availability Statement:** Data and codes from this study are available at: https://github.com/BanerjeeLab/CortexPattern.

**Funding:** MFS received no specific funding for this work. SB received funding from the Royal Society

## Abstract

The actin cortex is an active adaptive material, embedded with complex regulatory networks that can sense, generate, and transmit mechanical forces. The cortex exhibits a wide range of dynamic behaviours, from generating pulsatory contractions and travelling waves to forming organised structures. Despite the progress in characterising the biochemical and mechanical components of the actin cortex, the emergent dynamics of this mechanochemical system is poorly understood. Here we develop a reaction-diffusion model for the RhoA signalling network, the upstream regulator for actomyosin assembly and contractility, coupled to an active actomyosin gel, to investigate how the interplay between chemical signalling and mechanical forces regulates stresses and patterns in the cortex. We demonstrate that mechanochemical feedback in the cortex acts to destabilise homogeneous states and robustly generate pulsatile contractions. By tuning active stress in the system, we show that the cortex can generate propagating contraction pulses, form network structures, or exhibit topological turbulence.

## Author summary

The cellular actin cortex is a dynamic sub-membranous network of filamentous actin, myosin motors, and other accessory proteins that regulates the ability of cells to maintain or change shapes. While the key molecular components and mechanical properties of the actin cortex have been characterized, the ways in which biochemical signalling and mechanical forces interact to regulate cortex behaviours remain poorly understood. In this article, we develop a mathematical model for the actomyosin cortex that combines the reaction-diffusion dynamics of signalling proteins with active force generation by actomyosin networks. Using this model, we investigate how the feedback between mechanics and biochemical signalling regulates the propagation of actomyosin flows, mechanical stresses, and pattern formation in the cortex. Our work reveals a variety of ways in which the cortex can tune the dynamic coupling between biochemical activity, force production, and advective transport to control mechanical behaviours.

(URF/R1/180187), Human Frontiers Science Program (RGY0073/2018) and the National Institutes of Health (R35GM143042). EM received funding from the National Institutes of Health (R01GM098441) The funders had no role in study design, data collection and analysis, decision to publish, or preparation of the manuscript.

**Competing interests:** The authors have declared that no competing interests exist.

# Introduction

The actin cortex is an adaptive active material that dynamically regulates its mechanical properties to maintain or change cell shapes [1–4]. The actin cortex can display a wide range of dynamic behaviours from driving intracellular pulsatory contractions [5, 6] to cellular-scale polarized flows [5, 7, 8] and assembling protrusive or contractile structures during cell motility and cytokinesis [9, 10]. These behaviours must adapt to the cell's local environment and developmental stage. For instance, cellular-scale pulsatile contractions are often observed during developmental morphogenesis, where pulsatile contractions act as a mechanical ratchet to sequentially alter cell size and shapes [11–14], leading to tissue bending or elongation [15]. At the intracellular level, actomyosin pulses occur with chaotic spatiotemporal dynamics [5, 16, 17]. Travelling waves of actomyosin contraction can propagate across the cortical surface [16, 18, 20], and can be highly organised as in the case of surface contraction waves, which propagate as single waves from one pole of the cell to the other [19, 20]. In other physiological contexts, stable contractile structures are needed, as in the formation of stress fibers during cell adhesion [10], assembly of actomyosin purse-string during wound healing or gap closure [21–23], or the formation of a contractile ring during cytokinesis [24, 25]. While the biochemical pathways underlying actomyosin dynamics are known, the mechanisms by which actomyosin-driven mechanical forces feedback to upstream chemical signals to govern patterns and flows in the actin cortex remain poorly understood [4].

Alan Turing, in his seminal paper, demonstrated that reaction-diffusion systems can autonomously generate a wide variety of spatiotemporal patterns observed in nature, but noted that mechanical forces may also play an important role in pattern formation [26]. In recent years, purely biochemical models have been proposed for actomyosin pattern formation. RhoA and its downstream effectors of actin and myosin have been shown to form an activator-inhibitor system that exhibits excitable dynamics and oscillations [17, 27–29]. Here autocatalytic production of the activator RhoA leads to delayed accumulation of the inhibitor F-actin, producing local pulses of RhoA activity. Since F-actin diffusion is negligible compared to RhoA, this system produces travelling pulses of RhoA activity but cannot generate Turing patterns [27, 28]. By tuning RhoA production rates locally, static patterns of RhoA can be generated [28]. This raises the question if actomyosin patterns in the cortex strictly rely on biochemical cues or can spontaneously and robustly emerge via interplay between mechanical forces and biochemical signalling.

Mechanochemical feedback in the cytoskeleton is another mechanism for generating spatiotemporal patterns [30–34]. Active gel models of the cytoskeleton have suggested that pulsatory patterns can emerge from contractile instabilities driven by a positive feedback, in which active stress drives advective flows of stress producing factors such as actin and myosin [33, 35, 36]. These contractile instabilities cluster myosin into a few high concentration regions, but regulating myosin contraction with an independent RhoA oscillator can prevent the collapse of myosin and sustain oscillations [18]. However recent studies suggest a negative feedback loop between actomyosin and RhoA [17, 27–29], raising the question of how the feedback between RhoA and actomyosin mechanics is regulated to generate flows and patterns. Additionally, it remains unclear how the same molecular system can regulate the formation of stable contractile structures [4, 37] or exhibit turbulent dynamics [16].

Here we develop a reaction-diffusion model for an excitable signalling network comprising RhoA and actomyosin, coupled to the mechanics of an active gel representing the actomyosin network, to study how the feedback between biochemical signalling and mechanical stresses regulate mechanochemical patterns and flows in the actin cortex. Our mechanochemical model coupling RhoA signalling and transport with actomyosin mechanics goes beyond

previous theoretical models that are either purely biochemical [17, 27–29] or based on the mechanics of active gels [18, 33, 35, 36, 38]. Using this mechanochemical model, we ask how pulsatory flows arise from the coupling between a fast diffusing activator (RhoA) and a slow diffusing inhibitor (actomyosin), how mechanochemical feedbacks stabilise contractile instabilities into localised patterns, and how active stresses propagate local contractile signals or drive turbulent dynamics. Our model builds upon recent experimental observations of an activator-inhibitor relationship between RhoA and actomyosin [17], by introducing myosin-driven contraction to provide a mechanical feedback. By tuning just two biologically relevant parameters, the basal rate of RhoA production and the magnitude of active contractile stress, our model is able to capture a wide range of dynamic phases observed in the actomyosin networks, from travelling waves and pulsatile contractions to stable contractile networks and turbulent dynamics. We find that mechanochemical feedback acts to destabilise stationary states to robustly generate pulsed contractions. At high contractile activity and low basal rates of RhoA production, stable patterns of actomyosin emerge. Furthermore, the mechanochemical system encodes memory of transient perturbations that allows local signals to be translated into propagating contraction waves or stable patterns.

## Results

### Mechanochemical feedback generates robust pulsatile contractions

To elucidate the role of mechanochemical feedback in the generation of dynamic behaviours and instabilities in the actin cortex, we first study an ordinary differential equation model for the coupling between RhoA, actomyosin, and mechanical strain (Fig 1A). Here, for simplicity, we neglect spatial variations in chemical concentrations and study the coupled dynamics of RhoA and actomyosin in a locally homogeneous region of the cortex. Recent experiments on *Xenopus* oocytes [27] and *C. elegans* embryos [17] suggest that actomyosin pulsation in the cortex is regulated by the excitable dynamics of RhoA GTPase—the upstream regulator of actomyosin assembly and force production. Local autocatalytic activation of RhoA drives rapid initiation of RhoA pulses, followed by F-actin assembly and myosin recruitment. As actomyosin concentrations increase, F-actin dependent accumulation of the RhoA GTPase-activating proteins RGA-3/4 terminate the pulse through a delayed negative feedback [17].

Here, for simplicity, we represent F-actin and myosin as a single species—actomyosin—whose properties combine active force production (by myosin) and inhibition of RhoA (by F-actin). This representation is justified since the time course of appearance and disappearance of F-actin and myosin intensities measured locally during pulses of RhoA activity in the *C. elegans* cortex are remarkably similar, and the cortical lifetimes of F-actin and myosin II measured by single molecule imaging are also remarkably similar [17]. Based on experimental observations [17], we make the following assumptions in our model. First, RhoA is activated at a constant basal rate, $S$. Second, active RhoA promotes further production of active RhoA via an autocatalytic feedback, and also promotes the production of actomyosin. Third, actomyosin (F-actin) promotes local inactivation of RhoA by recruiting GAPs. With these assumptions, the rate of production of RhoA-GTP can be written in terms of the RhoA-GTP concentration $r$ and actomyosin concentration $m$:

$$R_r(r, m) = S + a\frac{r}{r_a + r} - g\frac{mr}{r_g + r}, \tag{1}$$

where $S$ is the applied stimulus (or basal rate of RhoA production) representing the activity of Rho-GEF, $a$ is the rate of autocatalytic production of RhoA, and $g$ is a negative feedback parameter arising from F-actin-driven accumulation of the GAP RGA-3/4 that inactivates

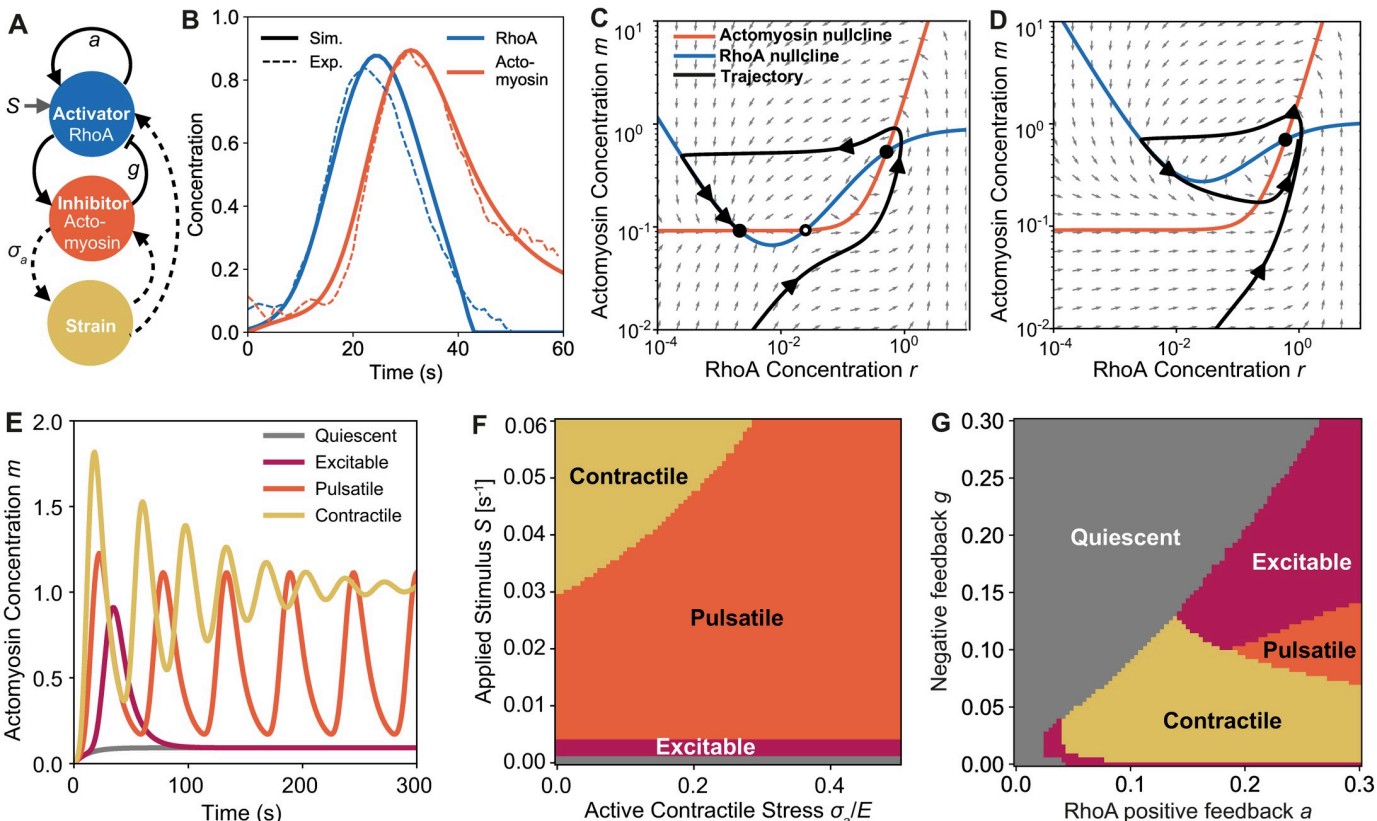

**Fig 1. Mechanical feedback sustains pulsatile contractions in excitable active medium.** (A) Feedback loop schematic of the system. Solid lines indicate biochemical feedback. Dashed lines indicate mechanical feedback. (B) RhoA (blue) and actomyosin concentration over time in the excitable phase during a single pulse. Dashed lines indicated experimentally measured values, solid lines show model with best fit parameters. (C-D) Trajectory curves and nullclines showing the fixed points for RhoA (blue) and actomyosin (orange) in the excitable (C) and in the pulsatile regime (D). Black arrows on the trajectory are equally spaced in time. Grey arrows show motion in phase space. Solid circles represent stable fixed points and the open circle is the unstable fixed point. (E) Concentration of actomyosin over time in the quiescent phase ($S = 0$), excitable phase ($S = 0.002$ $s^{-1}$), pulsatile phase ($S = 0.025$ $s^{-1}$), and contractile phase ($S = 0.075$ $s^{-1}$). (F) Phase diagram of the system, for varying contractile stress, $\sigma_a/E$, and applied stimulus, $S$, with $\eta_L/E = 5s$. (G) Phase diagram of the system, for varying rate of autocatalytic production of RhoA, $a$, negative feedback parameter, $g$. See Table 1 for a list of default parameters in the model.

RhoA. The constants $r_a$ and $r_g$ represent the threshold concentrations of RhoA above which the rate of RhoA production is independent of RhoA concentration. The rate of actomyosin production is given by:

$$R_m(r, m) = S_m + k_a r^2 - k_d m, \qquad (2)$$

where $S_m$ is the basal rate of actomyosin production, $k_a$ is the actomyosin assembly rate and $k_d$ is the disassembly rate. While the model parameters $S$, $S_m$, $k_a$ and $k_d$ are directly available from single-molecule data [17], the rest are calibrated by fitting our model to the experimental data for the concentration of RhoA and actomyosin during one contraction pulse (Fig 1B). The overall architecture of the biochemical circuit, namely autocatalytic production of RhoA and delayed feedback inhibition, is similar to recently proposed models for cortical RhoA dynamics [27, 29].

To introduce mechanical feedback into this model, we describe a locally homogeneous region of the cortex as an active viscoelastic material with strain $u$, with contractile stresses

generated by the actomyosin:

$$\eta_L \dot{u} + Eu = -\sigma_a \frac{m}{m_0 + m},$$

(3)

where $E$ is the compressional elastic modulus, $\eta_L$ is the local viscosity of the cytoskeletal element, $\sigma_a \, (> 0)$ is the maximum active stress arising from actomyosin-driven contractions, and $m_0$ is the concentration of actomyosin at half-maximum stress. The viscoelastic time-scale, $\tau = \eta_L/E$ and relative contractility $\sigma_a/E$ are estimated from available data on actin cortex of *C. elegans* [39]. Mechanical stress feeds back to the dynamics of both RhoA and actomyosin through conservation of mass; if the size of the system doubles then the concentration of chemical species must halve. Written explicitly, $\partial_u(c(1 + u)) = 0$, which implies that $\partial_u c = -c/(1 + u)$, where $c$ is the chemical concentration. While the contraction of a material in one or two dimensions should lead to an expansion in other dimensions, such deformation effects are not relevant for regulating the concentrations of membrane-bound RhoA and cortical actomyosin. It is likely that local contraction or dilation can drive variations in the thickness of cortical actomyosin, but how magnitude of cortical stress depends on thickness remains poorly understood and likely depends on details that vary across cells [40].

The governing equations for the coupled dynamics of RhoA and actomyosin are given by:

$$\dot{r} = R_r(r, m) - r\frac{\dot{u}}{1 + u} \, ,$$

(4)

$$\dot{m} = R_m(r, m) - m\frac{\dot{u}}{1 + u}.$$

(5)

In the above equations, actomyosin provides additional feedback to both RhoA and itself through changes in mechanical strain. An increase in local actomyosin concentration induces a local contraction, which in turn increases the actomyosin concentrations. By contrast, when actomyosin concentration decreases, there is local strain relaxation leading to a decrease in actomyosin concentration. Using the parameters calibrated from experiments, we simulated these dynamics numerically [41], for different values of the activity parameters $S$ and $\sigma_a$, to understand the role of mechanochemical feedback in actin cortex dynamics.

The model yields four distinct dynamic behaviours—quiescent, excitable, pulsatile, and contractile, depending on the magnitude of the applied stimulus, $S$ (S1 Fig). For low $S$, the system displays an excitable behaviour—a single pulse of RhoA, followed by a pulse of actomyosin and contractile strain buildup, before reaching a steady-state (Fig 1E). This excitable behaviour can be visualised in phase space as a trajectory about the intersecting nullclines (Fig 1C). For this excitable pulse, we observed a loop about the high $(r, m)$ fixed point before settling to a steady state at the lower fixed point.

As the applied stimulus is increased, other behaviours emerge (Fig 1E and 1F). For a small increase in $S$, we observe sustained pulsatile contractions, when the RhoA nullcline shifts up (Fig 1D), resulting in a single fixed point at high concentrations and the system is trapped in a limit cycle. At even higher values of $S$, the limit cycle is unstable and the pulse amplitude decays until the system settles in a contracted state with high actomyosin concentration and strain. Finally, for very low applied stimulus, we observe a quiescent mode; both RhoA and actomyosin steadily increase to a fixed set-point.

Previous work has shown that changes in concentration due to mechanical contraction can trigger oscillations in systems that otherwise remain stationary [42]. While pulsatile contractions are observed in the absence of contractile stress ($\sigma_a = 0$), we find that including mechanical feedback helps to sustain oscillations over a wider range of the parameter space (Fig 1F). As

actomyosin concentration drops after a pulse, the mechanical strain relaxes, further reducing both actomyosin and RhoA concentrations away from a fixed point, allowing another pulse to occur. These pulsatile states occur above a threshold value for the autocatalytic positive feedback parameter $a$ and for moderate values of the negative feedback parameter $g$ (Fig 1G). When the negative feedback is higher than the positive feedback we observe quiescent behaviour, where any changes in RhoA concentration are quickly slowed down by the negative feedback and brought to its equilibrium value. When $a$ is high compared to $g$ we observed several distinct regions of dynamic behaviours. When both $a$ and $g$ are low, the system settles to a contractile steady-state. When both $a$ and $g$ are high the system is excitable, and when $a$ is high and $g$ is moderate the system becomes pulsatile (Fig 1G).

## Spatial propagation of pulsatile flows and contractile pattern formation

To investigate the role of active mechanical stresses and RhoA signalling on spatial patterns and flows within the actin cortex, we develop a continuum active gel model of the actin cortex, coupled to the excitable RhoA signalling network and a viscous cytosol (Fig 1A). Using the *C. elegans* embryo as a model system, we develop a one-dimensional description of the actin cortex with periodic boundary conditions, assuming azimuthal symmetry around the long axis of the cell. The actin cortex is modelled as a porous Maxwell viscoelastic material, behaving elastically at short times and remodelling over longer time scales, with flows in the cortex advecting RhoA and actomyosin. The constitutive equation defining the time-evolution of stress field $\sigma$ $(x, t)$ is given by:

$$\tau \dot{\sigma} + \sigma = \eta \partial_x v + \sigma_a \left( \frac{m}{m_0 + m} \right) \tag{6}$$

where the spatial coordinate $x$ denotes distance along the surface of the cell, $\tau$ is the viscoelastic relaxation time scale, $\eta$ is the actomyosin network viscosity and $v(x, t)$ is the actomyosin flow velocity. The second term on the right hand side of the above equation is the active stress term with $\sigma_a$ the active contractile stress, and $m_0$ is the actomyosin concentration at half-maximum active stress. Local balance of viscoelastic forces with cytosolic drag, pressure and actomyosin-generated active contractile forces can be written as:

$$\Gamma(v - v_{\text{cyto}}) = \partial_x \sigma - \phi \partial_x P \,, \tag{7}$$

where $\Gamma$ is the frictional drag coefficient between the actomyosin network and the cytosol, $v_{\text{cyto}}$ is velocity of the cytosolic fluid, $P$ is the cytosolic pressure, and $\phi$ is the cytosol volume fraction. Conservation of mass implies, $\phi v_{\text{cyto}} = -(1 - \phi)v$. Darcy's law applied to the poroviscoelastic continuum yields a relationship between the pressure gradient and relative velocity between the cytosol and the cytoskeletal network [43, 44]

$$\Gamma \phi(v_{\text{cyto}} - v) = -\partial_x P \,. \tag{8}$$

By taking a spatial derivative in Eq (6), and using Eqs (7) and (8) alongwith mass conservation we obtain the following equation for the cytoskeletal velocity field:

$$\gamma(v + \tau \dot{v}) = \eta \partial_x^2 v + \sigma_a \partial_x \left( \frac{m}{m_0 + m} \right) \,, \tag{9}$$

where $\gamma = \Gamma(1 + \phi)/\phi$ is the effective drag coefficient. Mechanical feedback is then introduced through myosin-induced F-actin flows which advect both RhoA and myosin [18, 45], leading to their local accumulation due to convergent flows or depletion via divergent flows. These

advective flows compete with reaction and diffusion of RhoA and actomyosin:

$$\dot{r} + \partial_x(rv) = R_r(r, m) + D_r\partial_x^2 r, \tag{10}$$

$$\dot{m} + \partial_x(mv) = R_m(r, m) + D_m\partial_x^2 m, \tag{11}$$

where $D_r$ and $D_m$ are the diffusion coefficients for RhoA and actomyosin, taken from Nishi-kawa et al [18]. The default parameter for active stress and viscoelastic time scale are taken from Saha et al [39]. The model equations are then numerically integrated [46] in a periodic box of length $L = 10\lambda$, where $\lambda = \sqrt{\eta/\gamma}$ is the hydrodynamic length scale.

As shown in Fig 2A–2C, the numerical solutions predict a wide diversity of dynamic states as the active contractility $\sigma_a' = \sigma_a/\gamma$ and RhoA stimulus $S$ are varied– from stationary patterns to propagating waves and pulsatile flows (S2 Fig). At low $\sigma_a'$, diffusion dominates over advection, giving rise to spatially uniform concentration profiles that exhibit excitable, oscillatory, or contractile dynamics (Fig 2C, top row), as observed in the cellular-scale model (Fig 1). In the absence of mechanical feedback, reaction-diffusion alone cannot generate spatial patterns because the activator, RhoA, diffuses much faster than the inhibitor, actomyosin ($D_r \gg D_m$). In this regime, the homogeneous state becomes unstable and Turing patterns emerge, only when $D_r \ll D_m$ (S3 Fig).

As $\sigma_a'$ is increased, contractile instabilities develop due to local accumulation of actomyosin, allowing finite wavelength patterns to emerge. At low $S$ and high $\sigma_a'$, we observe stable localized peaks of actomyosin and RhoA (Fig 2C, bottom left). Initially, autocatalytic positive feedback of RhoA creates small RhoA concentration peaks (Fig 3A, 1st panel), which in turn produce actomyosin (Fig 3A, 2nd panel). As actomyosin begins to accumulate, large inward flows are generated, further increasing both RhoA and actomyosin concentrations (Fig 3A, 3rd panel). Finally, actomyosin-induced inhibition of RhoA results in RhoA localization on either side of the actomyosin peak (Fig 3A, 4th panel), in contrast to Turing patterns where activators and inhibitors overlap. With no RhoA advection, RhoA diffuses away from the peak, producing actomyosin behind it and generates a travelling wave (S4A Fig). With no actomyosin advection, actomyosin concentrations remain too low to prevent RhoA from diffusing and create a uniform steady state (S4B Fig). When several contractile actomyosin foci exist, they attract each other and merge into a single peak. This phase is reminiscent of equatorial RhoA zones during cell division [47, 48], and medial [49] and junctional [50] RhoA domains in polarized epithelia.

At higher $S$ and moderate $\sigma_a'$, we observe propagating waves and pulsatile flows (Fig 2C, bottom). A higher level of excitation in RhoA generates a localized actomyosin peak with high contractility. Away from the actomyosin peak, RhoA concentrations are higher and are advected towards actomyosin (Fig 3B, 1st panel). Advected RhoA produces actomyosin as it moves, such that the newly assembled actomyosin generates flows away from the centre, reducing the actomyosin concentration at the centre (Fig 3A, 2nd panel). Once the initial contraction dissipates, the two remaining actomyosin peaks merge, completing a cycle (Fig 3A, panels 3–4).

In the propagating waves state, we observe the periodic formation of RhoA pulses, which travel in waves before annihilating as two waves meet, as observed in the starfish oocyte [16, 27]. As $\sigma_a'$ is increased, actomyosin pulses generate large contractile flows that advect neighbouring pulses, creating chaotic, aperiodic motion. The pulses are highly localised, with small regions of high actomyosin concentration that form before dispersing, followed by a new pulse elsewhere, as in the *C. elegans* embryo [17]. Mechanical feedback through advection is necessary for the waves to form (S4D Fig). Without advection of RhoA however, we may still

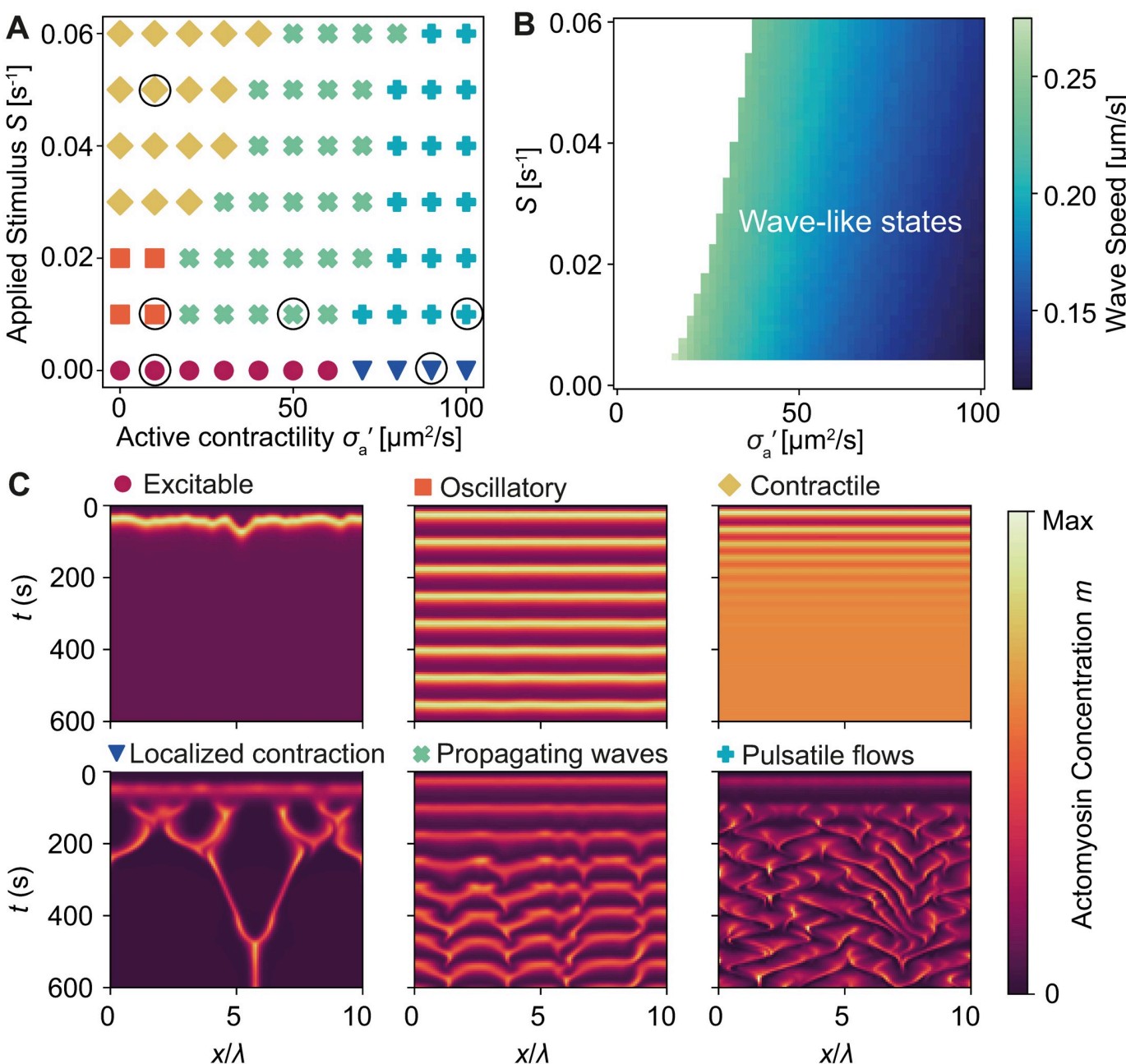

**Fig 2. Active stress generates spatial patterns and pulsatory flows.** (A) Phase diagram of the dynamic states of the system as a function of applied Rho stimulus $S$ and active contractility $\sigma'_a = \sigma_a/\gamma$. Encircled data points correspond to kymographs in panel C. (B) Actomyosin wave speed, computed from linear stability analysis of the model equations, for varying $S$ and $\sigma'_a$. (C) Kymograph of actomyosin concentration in different regimes (left to right, top to bottom): excitable, oscillatory, homogeneous contractile, localised contractions, propagating waves and pulsatile flows. See Tables 1 and 2 for a list of model parameters.

observe waves, since the actomyosin generated by RhoA forms clusters that travel with RhoA (S4C Fig). However, these waves display much less chaotic motion when compared to the system with advection (Fig 2C, bottom right).

Linear stability analysis of the continuum model reveals the role of active stress in destabilising the homogeneous state of the system (Fig 2B, S5 Fig). At low $S$, three fixed points exist (Fig 1B), with the lower fixed point being stable for high $\sigma'_a$. As $S$ is increased, the RhoA

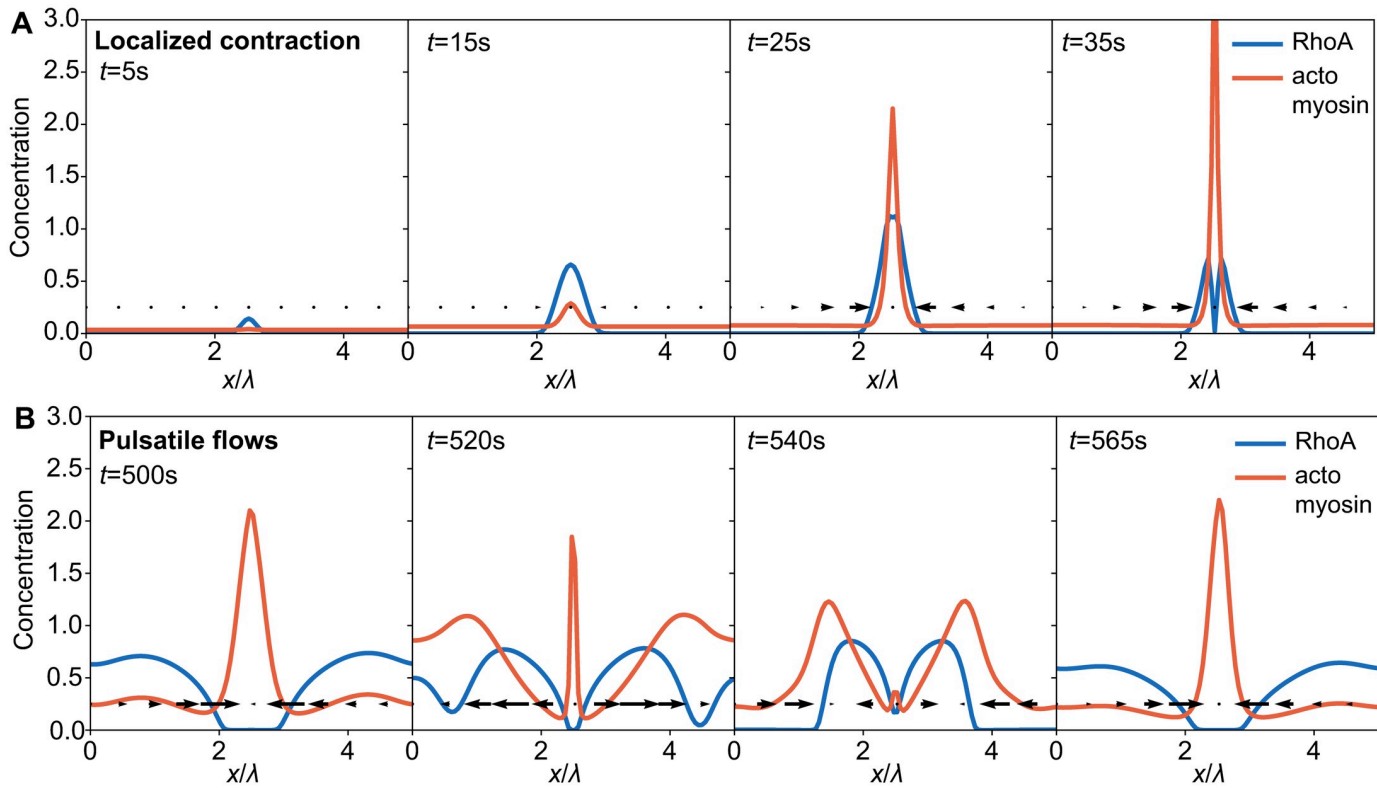

**Fig 3. Feedback mechanisms for pulsatile flows and stationary pattern formation.** (A) RhoA and actomyosin concentration during localized contraction ($\sigma'_a = 80\ \mu m^2/s$, $S = 0$) at t = 5s, 15s, 25s, and 35s (left to right). (B) RhoA and actomyosin concentration profiles over a pulsatile flow cycle ($\sigma'_a = 80\ \mu m^2/s$, $S = 0.01$ $s^{-1}$) at t = 500s, 520s, 540s, and 565s (left to right). Arrows indicate flow velocity.

nullcline shifts up until only one fixed point remains and the system enters the pulsatile regime (Fig 1C). Increasing $\sigma'_a$ leads to contractile instabilities that manifest as propagating waves and pulsatile flows (Fig 2C). While active stress is required for wave propagation, higher $\sigma'_a$ leads to lower wave speeds (Fig 2B).

## Response to local bursts of contractile activity

While our model can capture many of the dynamic states observed in the actin cortex, cells must have the ability to actively switch between flowing and contractile states during

**Table 1. Default biochemical parameters.**

| Parameter | Value |
|---|---|
| $S$ | $0\ s^{-1}$ |
| $a$ | $0.1609\ s^{-1}$ |
| $n$ | 1 |
| $r_a$ | 0.3833 |
| $g$ | $0.1787\ s^{-1}$ |
| $r_g$ | 0.01 |
| $S_m$ | $0.0076\ s^{-1}$ |
| $k_a$ | $0.1408\ s^{-1}$ |
| $k_d$ | $0.0828\ s^{-1}$ |

physiological transitions. Such state transitions may be triggered by a local up-regulation in RhoA activity, which may be induced in response to mechanical forces or by cell cycle checkpoints. But how mechanochemical coupling shapes the spatiotemporal response to transient localized changes in RhoA activity remains unclear.

To understand how RhoA signals can propagate through space and induce state transitions, we locally turned on RhoA activity and examined the output response. Starting from rest with no applied stimulus ($S = 0$), we applied an increased stimulus at the central region ($S = 0.03$ s$^{-1}$) (Fig 4A). By changing active stress in the system, we were able to regulate both the ability for the input RhoA to propagate in space, and for the system to remember the spatial location of the signal. We observed three distinct phases (Fig 4A): (i) propagation of a bistable front, where the memory of the signal location is lost and a global increase in RhoA concentration is observed, (ii) a soliton phase with transient spatial memory, and (iii) a high memory phase where a stable RhoA pattern is maintained.

At low active stress ($\sigma'_a = 12.5\ \mu\text{m}^2/\text{s}$), RhoA is excited into a pulsatile state within the activation region, while spreading laterally through diffusion (Fig 4A, left). A front of highly concentrated RhoA travels away from the source, increasing the total RhoA and actomyosin in the system (Fig 4B). This is reminiscent of a bistable front propagation in classical excitable systems [50, 51], where the system switches to the high concentration stationary state (Fig 2C).

As active stress is increased, we find that mechanical feedback is able to tune the properties of the biochemical system away from classical excitable systems. At moderate active stress ($\sigma'_a = 25\ \mu\text{m}^2/\text{s}$), RhoA pulses spread out as two solitons before annihilating as they meet (Fig 4A, middle), as seen in surface contraction waves [19, 20]. In contrast to classical excitable systems, actomyosin generated behind the RhoA wavefront increases its own concentration through contractile flows. This region of highly concentrated actomyosin behind the wave acts as a barrier that inhibits RhoA, preventing the system from switching to the high fixed point, and instead creates a soliton. Such a barrier can be seen at $\sigma'_a = 12\ \mu^2/\text{s}$, although it is too weak to prevent diffusion, with the band of reduced RhoA concentration behind the wavefront (Fig 4A, left).

At high active stress ($\sigma'_a = 50\ \mu\text{m}^2/\text{s}$), the contractile forces within the activated region are strong enough to maintain a spatially localized state that persists after the activation, remaining stationary in a fixed location at the centre of activation (Fig 4A, right). This suggests a potential mechanism for cells to direct the locations of contractility. Without the guidance of externally induced RhoA activation, the system in this parameter regime is incapable of spontaneously forming a stable pattern.

To quantify the input-output relationship of the system, we measured the correlation between the input signal and the output RhoA concentration at long times (Fig 4C). At low $\sigma'_a$, RhoA spreads outwards, leading to a loss of correlation between input and the output signal, akin to a memoryless system. For the soliton case, the shape of the input signal is remembered, resulting in a negative correlation as the waves travel away from the source. Finally, in the spatially patterned phase, a high-memory state emerges where RhoA remains localized at the centre of activation, with a strong positive correlation between the input and the output. These results suggest that mechanical stresses can play an important role in biochemical signal propagation, and in retaining the spatial memory of activity. For low active stress, signals propagate the fastest with global changes in contractility and no memory of the spatial location of the signal. As active stress is increased, we observe contraction waves propagating away from the source, displaying transient memory. At higher stresses, a high memory state develops, where transient local RhoA activations create localized contractile states.

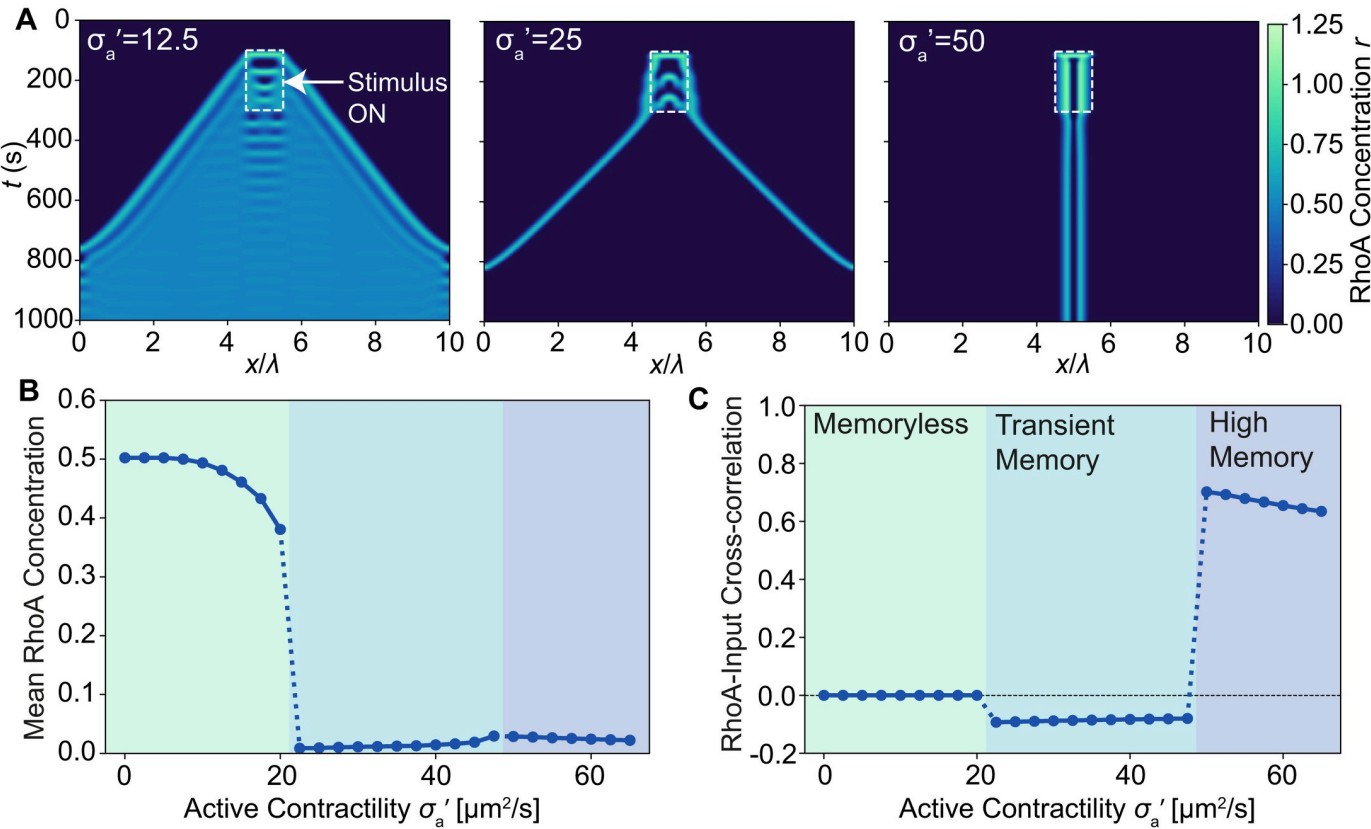

**Fig 4. Response to local bursts of contractile activity.** (A) Kymograph of Rho concentration upon local transient application of Rho stimulus: $S = 0.03 \text{ s}^{-1}$ inside box (dashed rectangle) and $S = 0$ outside box, for different values of active stress: (left) bistable front propagation for $\sigma'_a = 12.5 \ \mu m^2/s$, (middle) solitons for $\sigma'_a = 25 \ \mu m^2/s$, and (right) localised contraction for $\sigma'_a = 50 \ \mu m^2/s$. (B) Spatially averaged RhoA concentration, 300s after the application of stimulus. (C) Correlation between RhoA concentration and input stimulus $S(x, t)$, averaged over the last 120s.

## Network formation, pulsatile flows and topological turbulence

We now proceed to analyze our model in two spatial dimensions to investigate how contractile patterns form and propagate on the surface of a cell. The constitutive equation defining the time-evolution of the stress field $\sigma(\mathbf{x}, t)$ is given by

$$\tau \dot{\sigma}_{ij} + \sigma_{ij} = \eta_b \partial_k v_k \delta_{ij} + \eta_s (\partial_i v_j + \partial_j v_i - \partial_k v_k \delta_{ij}) + \sigma_a \left( \frac{m}{m_0 + m} \right) \delta_{ij} \qquad (12)$$

where $\mathbf{v}$ is the velocity vector, $\eta_s$ and $\eta_b$ are the shear and bulk viscosities, and we have assumed that the bulk and shear components have the same relaxation timescale, $\tau$. Using local force balance

$$\gamma v_i = \partial_j \sigma_{ij} \qquad (13)$$

we find that the equation governing the temporal evolution of the velocity field $\mathbf{v}$ is given by

$$\gamma (\mathbf{v} + \tau \dot{\mathbf{v}}) = \eta_s \nabla^2 \mathbf{v} + \eta_b \nabla (\nabla \cdot \mathbf{v}) + \sigma_a \nabla \left( \frac{m}{m_0 + m} \right). \qquad (14)$$

We assume that the bulk viscosity is higher than the shear viscosity, such that $\eta_b = \frac{3}{4}\eta$ and

$\eta_s = \frac{1}{4}\eta$, where $\eta$ is the experimentally measured viscosity [18]. The reaction-diffusion equations for RhoA and actomyosin concentration fields are given by

$$\dot{r} + \nabla \cdot (r\mathbf{v}) = R_r(r, m) + D_r \nabla^2 r, \tag{15}$$

$$\dot{m} + \nabla \cdot (m\mathbf{v}) = R_m(r, m) + D_m \nabla^2 m. \tag{16}$$

In the two-dimensional model, as we vary active stress ($\sigma_a$) and the applied stimulus of RhoA ($S$) we observe similar dynamic phases as in the one-dimensional model, though with more varied spatial organisation. At low applied stimulus and high active stress, we observe the formation of stable contractile networks (Fig 5A and S1 Video), analogous to the pattern formation phase observed in the one-dimensional model. Here, an initial excitable pulse forms many small cluster of highly concentrated actomyosin (Fig 5A, left panel). Over time, the clusters stretch and merge with other nearby clusters (Fig 5A, middle panel), eventually creating a space-spanning network of actomyosin, which gradually condenses into a stable configuration of fully connected edges and vertices (Fig 5A, right panel). These condensates of actomyosin are stabilized by contractile flows, similar to the localized contraction peaks observed in the one-dimensional model (Fig 3A). This sequence of actomyosin patterning is reminiscent of apical microridge formation in some epithelial cells [52].

As the applied stimulus $S$ is increased we observe pulsatile contractions and propagating waves of RhoA and actomyosin. At moderate active stress we observe propagating waves (Fig 5B and S2 Video), in which waves of actomyosin propagate with constant velocity until two waves collide and annihilate, with new waves periodically produced. Finally, at high applied stimulus and high active stress, we observe the pulsatile contraction phase (Fig 5C and S3 Video). Pulses of actomyosin contract into highly concentrated foci before dispersing and new pulses are formed. These dynamics are distinct from propagating waves as the pulses of actomyosin move more erratically as clusters merge, and show much more spatial variation in concentration. The corresponding velocity and strain fields are shown in S6 Fig.

The propagating waves of RhoA show features of topological turbulence, as recently reported in both the *Xenopus* eggs and embryo [27] and the starfish oocyte [16]. In this regime, RhoA and actomyosin exist with a limit cycle (Fig 1D), allowing us to extract the phase of oscillation using the relation $\phi = \tan^{-1}\left(\frac{r - \bar{r}}{m - \bar{m}}\right)$, where $\bar{r}$ and $\bar{m}$ are the mean RhoA and actomyosin concentrations (Fig 6A). We find that numerous topological defects exist in the phase field, indicating locations where the phase field is discontinuous, as reported in experiments [16, 27].

These may be parameterised by the winding number: the total amount that the phase changes by following an anti-clockwise contour around the defect. From the phase velocity field $\mathbf{v}_\phi = \nabla\phi$ (Fig 6B), the integer winding number is calculated as $\frac{1}{2\pi}\oint_C \mathbf{v}_\phi \cdot d\mathbf{s}$, where $C$ is a closed curve around the defect containing no others. In a +1 defect, the phase field spirals anti-clockwise about the defect, while in a -1 defect it spirals in a clockwise direction (Fig 6D). In our model, we find approximately the same number of +1 and -1 defects. In addition, oppositely charged defects attract one another, and annihilate when close enough, conserving the overall charge of the system (S4 Video).

The phase velocity also describes an effective velocity field due to a potential flow with point vortices of magnitude $\omega = \nabla \times \mathbf{v}_\phi$, with +1 and -1 defects corresponding to the center of vorticies with clockwise or anticlockwise rotations. We find a linear scaling between the effective kinetic energy $\bar{E} = \frac{1}{2}\langle \mathbf{v}_\phi^2 \rangle$ and the effective enstrophy $\bar{\Omega} = \frac{1}{2}\langle \omega^2 \rangle$, a measure of vorticity, for different values of active stress, in agreement with experimental data on RhoA in starfish oocytes [16]. Furthermore, when we quantify the phase velocity statistics, we find a power law

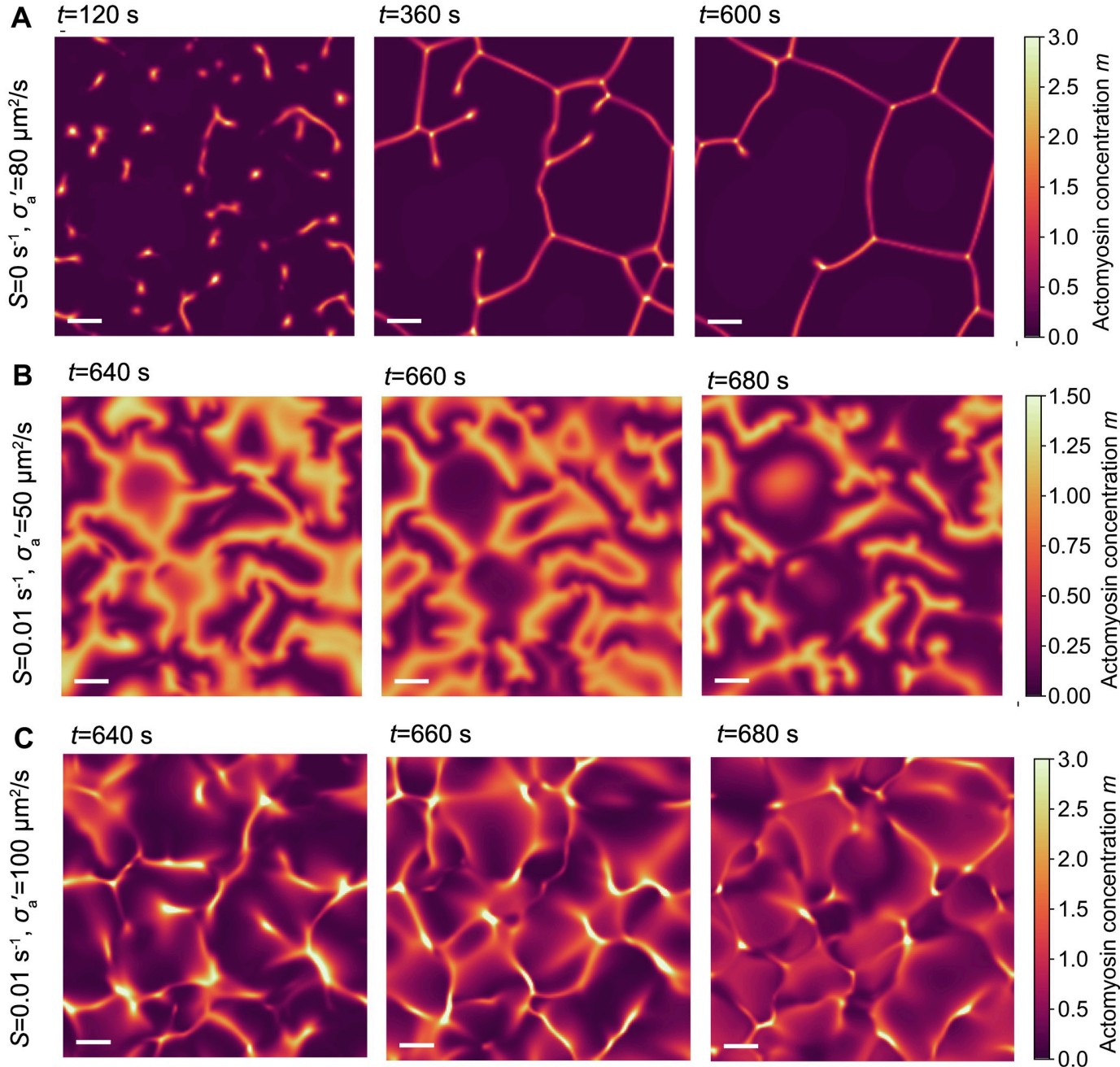

**Fig 5. Pattern formation and pulsatile flows in two spatial dimensions.** (A-C) Actomyosin concentration field obtained from simulations of the active gel model in two spatial dimensions, showing (A) contractile network formation ($\sigma'_a = 80 \ \mu m^2/s$, $S = 0$), (B) propagating waves ($\sigma'_a = 50 \ \mu m^2/s$, $S = 0.01 \ s^{-1}$) and (C) localized pulsatile contractions ($\sigma'_a = 100 \ \mu m^2/s$, $S = 0.01 \ s^{-1}$). Scale bar indicates a distance of $\lambda = 15 \ \mu m$.

tail in the probability density function which decays like $v^{-3}$, likely due to interacting vortices, consistent with experimental results [16] and theoretical predictions for vortex interactions [53]. Thus our model can quantitatively capture the key statistics and scaling laws of defect mediated turbulence in the RhoA phase field.

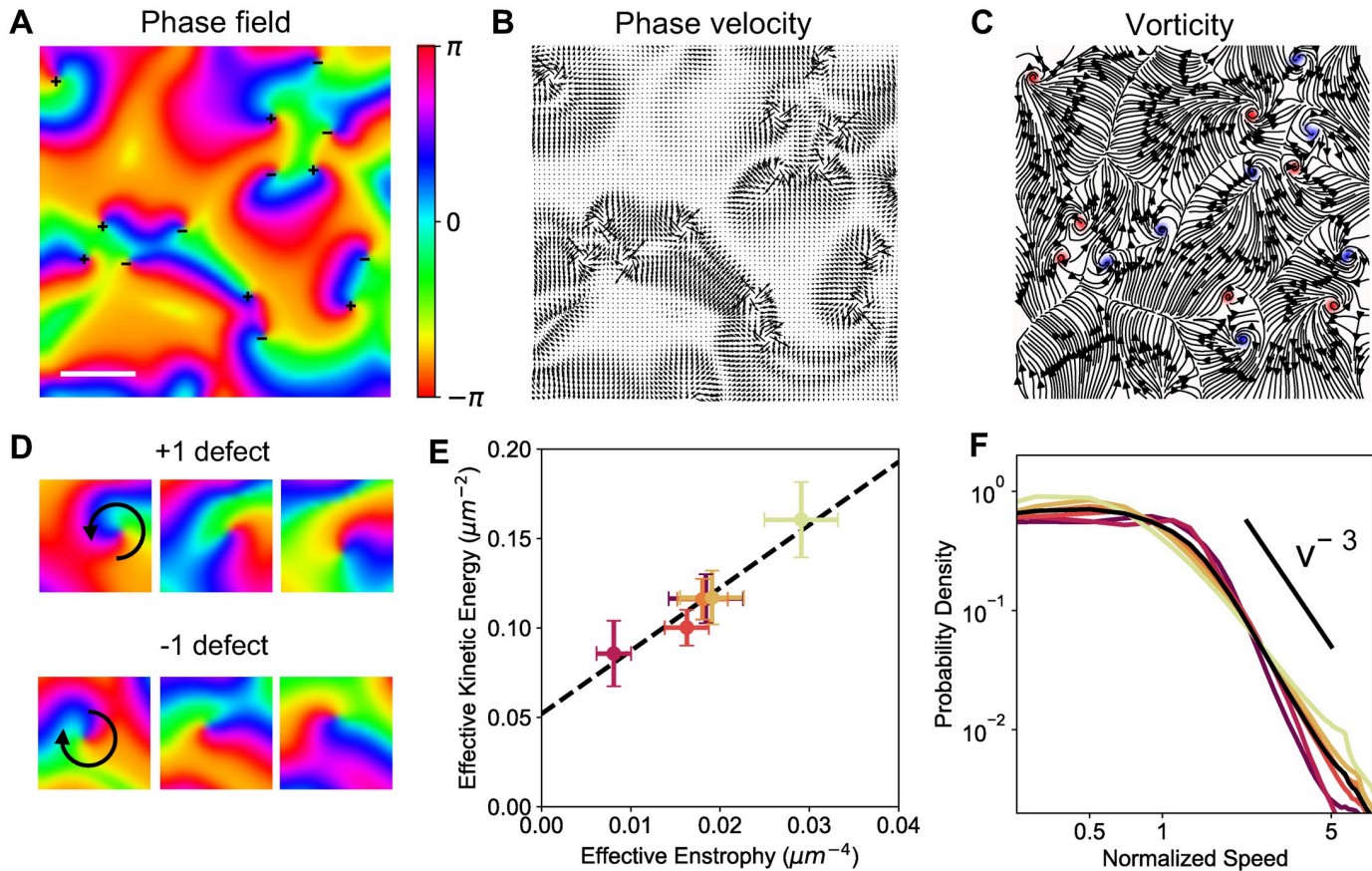

**Fig 6. Topological turbulence in RhoA flows.** (A) The RhoA phase field, $\phi$, corresponding to the dynamics in Fig 5B. Plus and minus symbols represent the location of +1 and -1 defects, respectively. Scale bar indicates a distance of $\lambda = 15\,\mu$m. (B) Spatial map of the phase velocity field $\mathbf{v}_\phi = \nabla\phi$. (C) Spatial map of contours of the phase vorticity field $\omega = \nabla \times \mathbf{v}_\phi$. Red indicates positive vorticity, corresponding to plus defects. Blue indicates negative vorticity, corresponding to minus defects. Lines indicate streamlines of the phase velocity field. (D) Representative images of a +1 defect (top) and -1 defect (bottom) over an oscillation cycle at (left to right) $t = 640$s, 660s, and 680s. (E) Linear scaling between the effective kinetic energy and the effective enstrophy of the RhoA phase velocity. (F) Probability density of normalized speed of RhoA phase field motion, for different values of active stress, $\sigma_a' = 50, 60, \ldots 100$. Light colours indicate higher magnitudes of active stress.

## Discussion

Activator-inhibitor coupling between RhoA and actomyosin has been shown to generate excitable dynamics and oscillations in the concentration of RhoA, F-actin and myosin-II [17, 18, 27, 29]. Purely biochemical feedback models with reaction-diffusion dynamics are thus sufficient to reproduce pulsatile dynamics of cortical RhoA and actomyosin concentrations as observed in experiments [17, 18, 27]. However, in contrast to Turing pattern-forming systems [26], the activator RhoA is fast-diffusing while the inhibitor actomyosin is slow-diffusing, thus the system should be expected to form spatially uniform states. Stochastic noise is therefore required to trigger the formation of waves in these excitable chemical systems without additional feedback mechanisms [17, 27, 29].

Mechanical forces generated by myosin-II in F-actin networks play an essential role in regulating the emergent dynamics of the cortical actomyosin networks. Mechanochemical feedback motifs in which mechanical strain locally upregulates actomyosin concentration through a mechanochemical coupling can lead to the formation of stress patterns [32, 54]. Alternatively, the contractile stresses generated by actomyosin drives advective flows of RhoA and

actomyosin up the concentration gradients, and thus providing an additional mechanical feedback through active stress. Active fluid models involving the diffusion and advection of active stress producing components, are cable of displaying both pulsatile and stable contractions without explicit coupling to signalling networks [33, 35]. In particular, it has been suggested that spontaneous pulsatory patterns may emerge when the fast-diffusing chemical species up-regulates and the slow-diffusing species down-regulates the active stress, or the up-regulator of active stress turns over faster compared to the down-regulator [35]. These conclusions, however, stand in contrast to the known properties of cortical RhoA and actomyosin, where the both the fast-diffusing RhoA and slow-diffusing actomyosin up-regulate the active stress.

Recent work suggests that feedback between chemical signaling and mechanical forces are essential to regulate pulsatile actomyosin dynamics in the cellular actin cortex [18]. Without coupling to chemical signaling, a positive feedback between contractile stresses generated by actomyosin and advective flows can lead to a density instability in the cortex, leading to the formation of high density myosin regions. By regulating myosin assembly and disassembly by an independent oscillatory signal, the contractile instability can be controlled, generating a pulsatile pattern of myosin-driven contractions. While this model neglects feedback between excitable RhoA and actomyosin, it illustrates how the coupling between signaling and mechanics can lead to spontaneous formation and propagation of RhoA waves without noise. At the cellular scale, dilution effects due to contraction could be capable of inducing oscillations in otherwise stationary systems [42]. However, the role of myosin-induced forces on wave propagation may be system dependent. In the *Xenopus* embryo, the myosin inhibitor blebbistatin had no significant effect on the observed pulses [27], but in the starfish oocyte, applying blebbistatin prevented surface contraction waves from propagating [20].

In this work, we couple actomyosin mechanics and biochemistry with RhoA signalling in a unified theoretical framework, where mechanical stresses and chemical signalling feedback to each other, regulating the emergent dynamics of the cortex. With this mechanochemical model, we are able to reproduce waves, patterns and pulsatile dynamics observed in the actin cortex by changing only two control parameters, namely the amount of active stress, and the basal rate of RhoA production. We demonstrate that mechanical feedback helps to robustly sustain pulsatile contractions, and provides a new mechanism for spatial patterning of RhoA and myosin, distinct from classical Turing patterns [26]. Having an excitable cortex with tunable dynamics and mechanochemical feedback can be beneficial in a number of morphogenetic contexts. Sustained pulsed contractions can act as a mechanical ratchet to sequentially reduce cell area and drive tissue bending [11–14, 55, 56]. Mechanochemical feedback may be important in enabling robust control of this morphogenetic ratchet. Cells may also display excitable behaviours which are governed by mechanics, for example, with active contractions triggered by large mechanical strains on the cell, helping to prevent tissue rupture [57, 58].

At low RhoA activity and high contractile activity, myosin forms stationary patterns, with RhoA localised on either side of the myosin peak. In two spatial dimensions, we observe the formation of many small clusters of high myosin concentrations, which stretch towards neighbouring clusters and condense into a stable contractile domains of actomyosin and RhoA. This is reminiscent of stable patterns observed *in vivo* on the apical surfaces of epithelial cells, including microridges [52], stable junctional and medial RhoA domains [49, 50]. As contractile activity is increased, the waves of myosin become highly peaked, creating pulsing contractile networks, reminiscent of pulsatile contractions in the *C. elegans* cortex [5]. At moderate activity, we observe periodic waves of myosin which annihilate before new wavefronts are generated in regions with low actomyosin concentration. These waves show features of topological turbulence, reminiscent of topological turbulence in RhoA concentration waves observed in *Xenopus* and starfish eggs [16, 27]. Besides generating patterns and flows, the level of active

contractile stress regulates the system's response to local RhoA signals, enabling phase transitions and memory entrainment as the activity is increased. Together, these results highlight the importance of considering both mechanical forces and chemical reactions when modelling the actin cortex, and they reveal a variety of ways in which cells can tune the dynamic coupling between RhoA activity, force production, and advective transport to control morphogenetic behaviours.

## Methods

### Linear stability analysis

To understand how mechanical stress and biochemical stimulus work together to generate pulsatile behaviour and propagating waves, we perform linear stability analysis about the homogenous steady-state (or fixed point) to determine the stability of the state, and the onset of oscillatory instabilities. In particular, we use linear stability analysis to compute the frequency, wave length and wave speed of the wave-like states that emerge from perturbations about quiescent homogeneous state. A perturbation of the form $\tilde{y}e^{ikx}$ about the steady state, $(r, m, 0)$, results in the following Jacobian:

$$\tilde{J} = \begin{pmatrix} \dfrac{aA}{(A+r)^2} - \dfrac{gmG}{(G+r)^2} - D_r k^2 & -\dfrac{gr}{(G+r)} & -ikr \\ 2k_a r & -k_d - D_m k^2 & -ikm \\ 0 & ik\dfrac{\sigma_a m_0}{\gamma\tau(m_0+m)^2} & -\dfrac{\gamma+vk^2}{\gamma\tau} \end{pmatrix}. \tag{17}$$

By finding the eigenvalue with the highest real part over all $k$, $\lambda(k^*)$, we obtain the fastest growing mode with wavelength $1/2\pi k^*$, frequency $\text{Im}(\lambda(k))$, and a wave speed of $\text{Im}(\lambda(k))/2\pi k^*$.

### Numerical method

The system of ordinary differential equations (Eqs 1–5) are numerically integrated using the SciPy package, which implements a Runge-Kutta 4(5) method. The 1D and 2D systems of partial differential equations are solved numerically using the FiPy python package, a finite volume PDE solver. The equations discretized in a periodic box with edge lengths $L = 10\lambda$, where $\lambda = \sqrt{\eta/\gamma}$ is the hydrodynamic length scale, using 150 grid points in each dimension. The system is then integrated forward in time using an implicity method, outputting in timesteps of $dt = 1$s.

### Model parameters

Tables 1 and 2 list the parameters used in our simulations. The RhoA-actomyosin chemical reaction model and parameters are obtained by fitting data from Michaux et al [17].

**Table 2. Default mechanical parameters.**

| Parameter | Value |
| --- | --- |
| $\tau$ | 5s |
| $\lambda$ | 14.3 $\mu$m |
| $\sigma_a/\gamma$ | 49.8 $\mu$m$^2$s$^{-1}$ |
| $D_r$ | 0.1 $\mu$m$^2$s$^{-1}$ |
| $D_m$ | 0.01 $\mu$m$^2$s$^{-1}$ |

Parameters for dimensionless active stress and viscoelastic time scale are taken from Saha et al [39], and myosin and RhoA diffusion coefficients are taken from Nishikawa et al [18].

## Supporting information

**S1 Fig. Dynamics in the activator-inhibitor mechanochemical model in response to changes in RhoA stimulus S.** (A-C) Dynamics of RhoA concentration (blue), actomyosin concentration (red), and contractile strain (yellow) in the (A) excitable phase ($\sigma_a/E = 0.2$, $S = 0.002$ s$^{-1}$), (B) pulsatile phase ($\sigma_a/E = 0.2$, $S = 0.025$ s$^{-1}$), and (C) the contractile phase ($\sigma_a/E = 0.2$, $S = 0.075$ s$^{-1}$).
(TIF)

**S2 Fig. Strain rate and RhoA kymographs in different phases of the continuum active gel model.** Kymographs of (A) strain rate ($\partial_x v$), and (B) RhoA concentration for the different phases in the active gel model corresponding to the actomyosin kymographs shown in Fig 2C.
(TIF)

**S3 Fig. Reaction-diffusion alone is not sufficient to generate pulsatile flows and waves in RhoA-myosin systems.** (A) Spatial wavelength, (B) temporal frequency, and (C) wave speed computed by linear stability analysis of the RhoA-myosin reaction-diffusion model with $\sigma'_a = 0$, for varying stimulus $S$ and activator to inhibitor diffusivity ratio, $D_r/D_m$. White regions in the parameter space show homogeneous steady states. Default parameter value of $D_r/D_m$ is 10. (D-E) Kymographs of actomyosin concentration (D) and RhoA concentration (E) showing pattern formation for low RhoA diffusivity ($D_r/D_m = 0.01$, $S = 0.01$ s$^{-1}$). (F) Steady-state spatial profiles of RhoA and actomyosin concentrations for the patterns corresponding to (D) and (E). Overlapping peaks indicates Turing pattern formation.
(TIF)

**S4 Fig. RhoA and actomyosin advection are necessary for stable pattern formation and propagating waves.** (A-D) Kymographs of actomyosin concentration with (A, C) no RhoA advection, and (B, D) no actomyosin advection. For (A, B) $S = 0.00$ s$^{-1}$, $\sigma'_a = 100$, and (C, D) $S = 0.01$ s$^{-1}$, $\sigma'_a = 100$.
(TIF)

**S5 Fig. Linear stability analysis predicts the frequency and wavelength of pulsatile contractions.** (A) Spatial wavelength, and (B) temporal frequency computed by linear stability analysis for varying active contractility $\sigma'_a$ and RhoA stimulus $S$.
(TIF)

**S6 Fig. Velocity field in two spatial dimensions.** (A-C) Actomyosin concentration field obtained from simulations of the active gel model in two spatial dimensions, showing (A) contractile network formation ($\sigma'_a = 80$ $\mu$m$^2$/s, $S = 0$), (B) propagating waves ($\sigma'_a = 50$ $\mu$m$^2$/s, $S = 0.01$ s$^{-1}$) and (C) localized pulsatile contractions ($\sigma'_a = 100$ $\mu$m$^2$/s, $S = 0.01$ s$^{-1}$). Scale bar indicates a distance of $\lambda = 15$ $\mu$m. Arrows indicate velocity field **v** and colour the local strain rate $\nabla \cdot \mathbf{v}$.
(TIF)

**S1 Video. Actomyosin network formation.**
(MP4)

**S2 Video. Propagating waves of actomyosin.**
(MP4)

**S3 Video. Pulsatile actomyosin contractions.**
(MP4)

**S4 Video. Topological turbulent flows of RhoA.**
(MP4)

## Author Contributions

**Conceptualization:** Michael F. Staddon, Edwin M. Munro, Shiladitya Banerjee.

**Data curation:** Michael F. Staddon, Edwin M. Munro.

**Formal analysis:** Michael F. Staddon, Shiladitya Banerjee.

**Funding acquisition:** Edwin M. Munro, Shiladitya Banerjee.

**Investigation:** Michael F. Staddon, Edwin M. Munro, Shiladitya Banerjee.

**Methodology:** Michael F. Staddon, Shiladitya Banerjee.

**Project administration:** Shiladitya Banerjee.

**Resources:** Shiladitya Banerjee.

**Software:** Michael F. Staddon.

**Supervision:** Edwin M. Munro, Shiladitya Banerjee.

**Validation:** Michael F. Staddon, Edwin M. Munro, Shiladitya Banerjee.

**Visualization:** Michael F. Staddon, Shiladitya Banerjee.

**Writing – original draft:** Michael F. Staddon, Shiladitya Banerjee.

**Writing – review & editing:** Michael F. Staddon, Edwin M. Munro, Shiladitya Banerjee.

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
