## [Decision Letter · Decision Letter 0]

16 Nov 2021

Dear Prof. Banerjee,

Thank you very much for submitting your manuscript "Pulsatile contractions and pattern formation in excitable actomyosin cortex" for consideration at PLOS Computational Biology.

As with all papers reviewed by the journal, your manuscript was reviewed by members of the editorial board and by several independent reviewers. In light of the reviews (below this email), we would like to invite the resubmission of a significantly-revised version that takes into account the reviewers' comments.

All 3 reviewers appreciated the study, but all have a number of critiques, some of them major,

which you will have to address. We cannot make any decision about publication until we have seen the revised manuscript and your response to the reviewers' comments. Your revised manuscript is also likely to be sent to at least one of the reviewers for further evaluation.

Sincerely,

Alex Mogilner

Guest Editor

PLOS Computational Biology

Mark Alber

Deputy Editor

PLOS Computational Biology

Dear Authors, all 3 reviewers appreciated the study, but all have a number of critiques, some of them major,

which you will have to address. I will send the revised manuscript to at least one of the reviewers for the

second look.

Reviewer's Responses to Questions

**Comments to the Authors:**

Reviewer #1: Review on the manuscript "Pulsatile contractions and pattern formation in excitable actomyosin cortex"

by M.F. Staddon, E.M. Munro and S. Banerjee submitted for publication to PlosCompBiol.

This is a modelling and simulation study motivated by recent experimental observations [17] on pulsed contractions in the cortex of C. Elegans. The authors formulate a mathematical model for the feedback between RhoA and Actomyosin which has both, a biochemical component (the know Activator-Inhibitor Loop between RhoA and actomyosin) and a mechanical component (the cytoskeleton flow in respone to actomyosin contraction).

The study highlights that

1) the activator-inhibitor system alone features a series of different steady state solutions, oscillating patterns and limit cycles depending on parameter values.

2) the spatially inhomogeneous version of this model coupled to contractile flow in one spatial dimension (periodic boundary cond.) features complex spatiotemporal patterns depending on paramters: either oscillating or non-oscillating paterns in time and convective or non-convective patterns in space, as well as any combination of these depending on parameter values.

3) the same observation is made for the model simulated for a 2-dimension spatial domain. The resulting patterns coincide qualitatively well with recent observations for patterns in the cortex of C. Elegans [17] (pulses) and starfish oocyte (propagating waves).

I appreciate that this is a very interesting study using an innovative and beautiful modelling approach which succeeds in explaining qualitatively some of the patterns observed in the cortices of various cell types. This alone I believe already deserves publication as I believe it will pave the way to even more intricate and powerful modelling of actomyosin contraction with realistic cell geometries.

The only concern I have is about the interpretation/parametrisation of the governing equations for the cytoskeleton flows. The derivation relates stress to drag in equ (7). The drag in this expression is taken as independent of the concentration of actomyosin m (the authors lump together the concentrations of F-Actin and of myosin into m). I'd argue that a more appropriate formulation of this relation should involve the density m at least as linear factor in the drag term (more actomyosin -> more drag and vice versa).

The way the authors write the equations allows them to derive the governing equation (8) without disturbing additional factors m and m_x, which would probably make it much harder to solve the resulting momentum equation (8) (I guess the velocity field would blow up where m is close to 0). I recognize that probably some sort of regularisation would be needed and omitting m from (7) appears to be the authors' approach of doing that. But that approach (assume the cytoskeleton density is constant) is inconsistent with the rest of the modelling, namely that for m a separate continuity equation (10) is stipulated and that according to simulations the value of m ranges between 0 (depending on parameters in most of the domain) and Max. I think all that has to do with not distinguishing between F-Actin concentration and myosin concentration and I believe that this has to be sorted out before publication so the model can be interpreted properly with regards to experimental observations.

On that note: it would be good a plot of the simulated velocity field could be added, e.g. to Figure 6 which only show the dynamics of the phase field so far, or to the SM.

Reviewer #2: The authors present a theoretical study of pattern formation in zero-, one- and two-dimensional systems of excitable actomyosin. The main scheme is presented in Fig. 1a and essentially based on experimental results from Ref. 17. The central element is RhoA, that is an activator because it promotes its own activity and governs many downstream targets. In contrast to most other work in this field, actomyosin is considered to be one entity only. Most importantly, there is negative feedback from actomyosin back to RhoA, through the GAP RGA-3/4, as described in Ref. 17 for C. elegans. Together, this biochemical system is similar to one of the activator-inhibitor systems envisioned by Turing. In a last step, it is coupled to the mechanics of the system. Here different models are discussed throughout the paper. First we have a very simple model where contractile strain leads to increase in concentration, thereby promoting RhoA and actomyosin. This model has no spatial domain. Later flow and advection are introduced, very similar to work from the Dresden group in Refs. 30 and 32, an considered for 1D and 2D. The authors report a large range of possible behaviour, including quiescence, contraction, pulsations, excitable excursions, solitons, chaos, turbulence. Phase diagrams are provided. The relation to experiments occurs mainly by several comments in the text pointing out similarities to observations in published work.

This work is rather theoretical in nature and in general is an interesting and original contribution to the large field of actomyosin as excitable medium. I especially like the nice progression from simple to more complex models throughout the course of the paper. Yet I also see two major weaknesses. First it is not entirely clear what really is new in regard to theory and how it compares to published work. Second the relation to experiments is unclear. Especially for PLOS Computational Biology (in contrast to a pure physics journal) one would like to see a stronger connection to experiments, especially when one of the authors is very well known for such experiments. Below I comment a bit more on both aspects.

Regarding theory work, I feel that much earlier work has to be cited and discussed. Regarding the scheme in Fig. a, the authors should also discuss Kamps, Dominic, et al. "Optogenetic Tuning Reveals Rho Amplification-Dependent Dynamics of a Cell Contraction Signal Network." Cell reports 33.9 (2020): 108467. Their biochemical model is rather similar but more detailed and therefore generates oscillations by itself, without the mechanics. Moreover the negative feedback from myosin to RhoA is another one than mentioned here (inhibition of GEF rather than upregulation of GAP). This leads to the question how specific the model here is to C. elegans and the particular GAP identified here, and which other models should be considered in general ? How would the authors’ results change when considered the model from Kamps et al. for their biochemistry part ?

My second comment is that the effect of concentration increase upon contraction is also not new and has been used for modelling e.g. in this paper: Buttenschön, Andreas, Yue Liu, and Leah Edelstein-Keshet. "Cell size, mechanical tension, and GTPase signaling in the single cell." Bulletin of mathematical biology 82.2 (2020): 1-33. Concentration effects are also often described in the context of cell growth, when volume is increasing. The authors should discuss better how their model compares with these other models. I also miss a discuss of the effect of three dimension: concentration might stay constant because due to the Poisson effect, contraction in one dimension comes with extension in another dimension. Is there experimental evidence that concentration effects are really relevant?

I am a bit confused about the mechanical model used. Eq. 3 looks like Kelvin-Voigt to me, but Eq. 6 is Maxwell. Why do the authors change the viscoelastic model and why is this not explained? Do the concentration effects of the 0D model also exist in the 1D and 2D model?

Regarding the phase plane analysis in Fig. 2, I wonder what happens to the third component u. The system is 3D and I do not understand how one can analyze this without the dynamics in u. Is there some kind of adiabatic approximation used here?

Recently there has been much work on excitable actomyosin coupled to shape changes, compare e.g. Brinkmann, Felix, et al. "Post-Turing tissue pattern formation: Advent of mechanochemistry." PLoS computational biology 14.7 (2018): e1006259 or the very recent work by the Turlier lab on "A viscous active shell theory of the cell cortex" (https://arxiv.org/abs/2110.12089). Maybe the authors want to comment on their model in this context.

I also want to comment that this statement here is not correct: “Travelling waves of actomyosin contraction can propagate across the cortical surface [16, 18].” Such a traveling wave has been described for starfish oocytes in Ref. 20, but not in Ref. 18, where it is explicitly shown that blebbistatin has no effects, so these waves are myosin-independent (in contrast to Ref. 20, where blebbistatin abolishes the wave). The model in that paper works without myosin and the negative feedback in the model is only through actin polymerization. The authors should consider how this fits into their framework and if e.g. the phase analysis for turbulence really depends on the specific model.

Finally I think that the authors should better explain the relation to experiments. In my view, they should include experimental data going beyond the two curves shown in Fig. 1b. For the bulk of the paper, there are only theoretical predictions and no direct comparison with experiments, although the authors make a large effort to parametrize with realistic values. For example, how do the correlation lengths from Fig. 5 and 6 correspond to structure formation in the C. elegans cortex? Actually this system shows many aster-like structures and it is not clear if they are predicted here. Can one compare theoretical and experimental flow fields, e.g. after local stimulation? Such comparison would strongly increase impact.

Reviewer #3: 1. The feedback loop of activator-inhibitor-strain (Fig.1) is of course a simplified network extracted from the more elaborate Rho-Rock network. Would the coarse graining implicit in this simplified network not lead to delay terms.

If so how would these affect the dynamics.

2. In the present study, the arrow from strain to Rho activation (Fig.1) arises from the dilution effect, \\partial_u c (1+u) = 0, doesn’t it ? Could there be explicit local strain sensing via signalling?

3. The flows that are discussed are cytoskeletal flows and not the cytosolic velocity that the system is embedded in. The cytoskeletal flows are therefore compressible. It would be good to clearly state this, as well as provide a justification

for how you have ignored the cytosolic velocity field. Does the dilution effect only apply to chemical species bound to the elastomer?

4. The dynamical transition of the front propagation of Rho from stationary to moving to diffusive might be associated with different front propagation dynamical exponents, and may be analysed using ideas from Van Sarloos seminal work.

5. In equation 11, shouldn’t the elastic response have both shear and compression? In which case the derivation of Eq. 11, would be more involved.

6. I did not see details of the numerical algorithm in the manuscript. Could this be provided please.

**Have the authors made all data and (if applicable) computational code underlying the findings in their manuscript fully available?**

Reviewer #1: Yes

Reviewer #2: Yes

Reviewer #3: **No: **The details of the numerical solutions need to be presented.

PLOS authors have the option to publish the peer review history of their article (what does this mean?). If published, this will include your full peer review and any attached files.

Reviewer #1: No

Reviewer #2: No

Reviewer #3: No
---

## [Decision Letter · Decision Letter 1]

1 Mar 2022

Dear Prof. Banerjee,

We are pleased to inform you that your manuscript 'Pulsatile contractions and pattern formation in excitable actomyosin cortex' has been provisionally accepted for publication in PLOS Computational Biology.

Best regards,

Alex Mogilner

Guest Editor

PLOS Computational Biology

Mark Alber

Deputy Editor

PLOS Computational Biology

Reviewer's Responses to Questions

**Comments to the Authors:**

Reviewer #2: The authors have responded very well to all reviewer comments. In particular, they now cite and discuss the papers by Kamps et al. and Buttenschön et al., which are very relevant here. They also explain well that this work is well grounded in experimental results (mainly for C. elegans) and discuss better the relation to known observations in different model organisms. The nature of the different mechanical models has been clarified and new work on poroviscoelasticity has been added. I congratulate the authors to this very nice work.

**Have the authors made all data and (if applicable) computational code underlying the findings in their manuscript fully available?**

Reviewer #2: Yes

PLOS authors have the option to publish the peer review history of their article (what does this mean?). If published, this will include your full peer review and any attached files.

Reviewer #2: **Yes: **Ulrich S. Schwarz

---

## [Editor Report · Acceptance letter]

25 Mar 2022

PCOMPBIOL-D-21-01832R1 

Pulsatile contractions and pattern formation in excitable actomyosin cortex

Dear Dr Banerjee,

I am pleased to inform you that your manuscript has been formally accepted for publication in PLOS Computational Biology. Your manuscript is now with our production department and you will be notified of the publication date in due course.

With kind regards,

Zsofia Freund
